# Beyond Contraception: Pharmacist Roles to Support Maternal Health

**DOI:** 10.3390/pharmacy10060163

**Published:** 2022-11-30

**Authors:** Natalie DiPietro Mager, David Bright, Allie Jo Shipman

**Affiliations:** 1Department of Pharmacy Practice, Raabe College of Pharmacy, Ohio Northern University, Ada, OH 45810, USA; 2Department of Pharmaceutical Sciences, Ferris State University College of Pharmacy, Big Rapids, MI 49307, USA; 3National Alliance of State Pharmacy Associations, Richmond, VA 23235, USA

**Keywords:** pharmacist, pharmacy practice, maternal health, community-based pharmacy, women’s health

## Abstract

While contraception prescribing by pharmacists has seen rapid growth in recent years, pharmacist-provided services that can impact maternal health encompass more than just contraception. Each phase of maternal health—preconception, pregnancy, and post-pregnancy—has unique needs, and pharmacists are well equipped to provide services to meet those needs and are more accessible than other healthcare providers. While pharmacist-provided maternal health services may lead to significant savings to the healthcare system, additional research to more fully capture the value of pharmacist-provided maternal health services is needed. Robust implementation of a pharmacist-provided maternal health services program will require partnerships between providers, payers, and pharmacists. Infant and maternal mortality, preterm birth, and unintended pregnancies are significant public health issues, and pharmacists should be seen as a capable workforce who can provide needed maternal health care and serve as a gateway into the healthcare system for those capable of pregnancy.

## 1. Introduction

Maternal health refers to the health of women during the pre-pregnancy, pregnancy, and postpartum periods [1]. The purpose of providing maternal health care is to improve the health of women, as well as the outcome of any pregnancy they may choose to have. In the United States (U.S.), maternal health is monitored and evaluated through assessment of the prevalence of chronic disease states and health behaviors among women of reproductive age as well as pregnancy-related morbidity and mortality [2]. Higher rates of chronic health conditions, obesity, and maternal morbidity and mortality along with higher rates of infant mortality than comparable countries demonstrate the need for provision of routine maternal health care in the U.S. [3].

When conceptualizing maternal health, it is important to recognize that there are a broad series of factors that can adversely impact health through one’s lifetime. These health burdens not only may adversely impact women of childbearing age themselves, but also may impact a pregnancy [2]. Waiting for a patient to receive prenatal care is likely too late to address many risk factors for adverse maternal-fetal outcomes, especially as nearly half of pregnancies in the U.S. are unintended [4]; further, 1 in 16 pregnant women in the U.S. receive no or inadequate prenatal care [5]. Ensuring access to affordable and comprehensive health care is a key component to improving health outcomes [6].

Currently, however, the U.S. health care system suffers from a shortage of primary care providers, including obstetricians and gynecologists, and financial barriers to care [6]. It is estimated 11% of women in the United States are uninsured, with millions more underinsured [7]. Women may not be able to access routine, preventive health care services or may delay needed care [6]. As part of the comprehensive health care team, pharmacists are well-positioned to close many of these gaps in maternal health care [8]. While much attention has been given in recent years to legislation that grants pharmacists prescriptive authority over contraception products, it must be made clear that pharmacy-based maternal health services do not stop at the provision of contraception.

The purpose of this perspective is to describe the broad impact pharmacists may have on the provision of maternal health services, as well as forward-looking considerations to accelerate service implementation and expansion. Within the context of this article, it should also be noted that not all people who become pregnant or give birth identify as women; therefore, while “women’s health” and related terms may exist in the body of literature and may continue to be used in this area in some publications referenced in this perspective, terms used throughout this perspective should be interpreted to imply inclusive and intentional care for all who are capable of becoming pregnant.

## 2. Pharmacists’ Potential to Fill Gaps in Care

Maternal health refers to health in the pre-pregnancy, pregnancy, and postpartum periods [1]. Patients need high quality healthcare services, education and support before, during, and after pregnancy as part of comprehensive primary care [9,10]. All healthcare providers and teams should be cognizant of the factors that contribute to overall health and should regularly identify and intervene on potential risks for reproductive-age patients (typically 15–45 years) [2]. Every patient interaction with a healthcare provider represents an opportunity to ensure that evidence-based standards of care as well as social needs are being met in a culturally appropriate manner [2,11]. Although usually framed as well-woman care, these services should be provided as appropriate to all patients capable of becoming pregnant regardless of gender [11].

About 96% of the total population lives within 10 miles of a community pharmacy [12]. Such accessibility can be helpful for patients that would otherwise travel long distances to see a physician. In addition, patients are much more likely to frequent a pharmacy than a physician’s office. Specific to the Medicare population, the median number of annual visits to community pharmacies is nearly double that which is observed among primary care physician office visits [13]. Across a more general population, the difference widens, with 35 visits to pharmacies as compared to 4 visits to primary care providers [14]. Therefore, pharmacists as providers of women’s and maternal health care services may be logical from a convenience, accessibility, and cost standpoint.

Pharmacists are trained to provide comprehensive, quality health care services. Studies have shown positive outcomes when pharmacists are involved in the management of chronic medical conditions, and immunization rates have improved since pharmacists have been authorized to administer vaccines [15]. Pharmacists are providing medical screening, health and wellness counseling, medication management services, and patient education. Primary care services, including women’s health, can be provided by pharmacists to help alleviate the shortage of primary care physicians [16]. 

Pharmacists are capable of providing many of the needed maternal health services and, given their accessibility, may fill a gap for patients not receiving these services elsewhere [8]. For a few of the services listed below, pharmacists will need to refer patients to other healthcare providers through clinical-community linkages [8]. In addition to prevention services, pharmacists play an important role in disease state management; chronic conditions such as hypertension and diabetes need to be well-controlled pre-pregnancy to optimize the health of patients and mitigate risks for adverse maternal-fetal outcomes [11]. Yet, a recent study showed that for non-pregnant women aged 20–44 years with hypertension, nearly 41% were uncontrolled and among those with diagnosed diabetes, nearly 52% were uncontrolled [17]. Furthermore, pharmacists are in a perfect position to review and assess a patient’s current medication profile to identify potentially teratogenic medications and supplements that may need to be discontinued prior to conception, and to ensure effective contraceptive coverage in the meantime to avoid inadvertent exposures [11].

## 3. Clinical and Service Considerations

Community pharmacists wanting to initiate such services should consult evidence-based resources to determine where to begin. Table 1 summarizes the current recommendations from the Women’s Preventive Services Initiative (WPSI) regarding well-woman care for reproductive-age women [18]; as these are updated annually, clinicians should ensure they are referencing the latest version of the guidelines [19]. For the purpose of this manuscript, the WPSI Recommendations are referenced, but it should be noted that additional resources have been produced by other organizations that may be helpful to pharmacists. For instance, the U.S. Centers for Disease Control and Prevention (CDC) also has several resources available around preventive, preconception, and pregnancy planning services that pharmacists can reference [20].

After reviewing Table 1, a pharmacist may realize that they already provide one or more of these services and can consider how they may want to frame and promote the services as related to maternal health, and which new services may make sense to add onto the existing ones. Similarly, if a community pharmacist is already providing direct access to contraceptive services or dispensing a contraceptive product, it may not be a stretch to add on other needed services as appropriate [21]. Pharmacists may find it helpful to consult outside resources to receive practical guidance on how to implement or expand a pharmacy service [22,23,24,25,26,27,28,29,30].

Figure 1 shows the recommended pharmacists’ maternal health services sets and provides a visualization for how the services relate and should be implemented. All patients, regardless of pregnancy status, should receive the Women’s Health Services Set [8]. Pregnant patients should receive the Pregnancy Services set as appropriate [8]. For non-pregnant patients of reproductive potential, pharmacists should assess ability or desire to conceive in the next year (also known as the “One Key Question Initiative^®”^) [8,11]. Patients not currently planning a pregnancy should be offered contraceptive services if desired and should also still receive education and counseling to reduce risks, should unintended pregnancy occur. Patients desiring a pregnancy should receive the Preconception (prior to first pregnancy) or Interconception (between pregnancy) Service Set in order to optimize health before pregnancy [8]. Once the recommended intervention has been performed, the pharmacist should follow-up as appropriate to ensure the patient receives any additional required services; for example, blood pressure screening may result in pharmacist-provided management of the disease state or a referral to a prescriber, per scope of practice.

Regardless of which service set is applied, pharmacists should not lose sight of care foundations that would apply to other conditions and populations. For example, when providing care, documentation and communication to ensure coordinated care is paramount. Enhanced interoperability of electronic health records, such as integration of community pharmacies into health information exchange, will facilitate care coordination and reduce fragmented care [2]. In addition, while there are multiple clinical considerations to be addressed, pharmacists must also consider social determinants of health (SDoH) that have a large impact for some patients [2,6,11]. Using the Pharmacists’ Patient Care Process, pharmacists can identify and address patient SDoH-related needs [31]. Through this, pharmacists may be able to help address disparities currently seen in maternal health based on geography or race/ethnicity [2].

## 4. Value of Interventions

The return on investment of maternal health services is difficult to quantify due to the limited studies on the topic, the varied nature of interventions, and the extended time horizon to see potential benefits [32,33]. In order to develop strong economic data, consensus and standardization of study approaches would be helpful [33]. Until greater clarity emerges on long-term economic impacts of interventions, other strategies, such as comparing cost per intervention when provided by a physician versus when provided by a pharmacist, may be useful in the near term. Such analyses may still be difficult to conduct as a physician may, for instance, prescribe a contraception product during another visit, with a total cost that accounts for multiple interventions. However, knowing that there are visits that are essentially exclusively for the provision of contraception prescriptions, simpler math may be done to suggest cost savings when pharmacists prescribe contraception products. While one could argue that pharmacist-provided services may reduce physician visits, which could reduce other cost-effective and beneficial screenings, recent studies do not suggest that this objection should be a hindrance to pharmacist-prescribed contraception [34]. In some cases, particularly in areas with a primary care provider shortage, leveraging pharmacists to provide this service may help with local health care capacity issues as well. Therefore, consideration of the direct and indirect costs of pharmacist-provided service provision may allow for simpler justification for selected pharmacist interventions. 

A study in Oregon, the first U.S. state to authorize pharmacists to independently prescribe hormonal contraceptives, showed significant benefit to the state Medicaid program when pharmacists were involved. Within the first two years of the policy change, pharmacists wrote 10% of new prescriptions for hormonal contraceptives, with 74% of those patients not using any form of birth control in the month prior to the pharmacist’s prescription. The study also showed that pharmacists prevented more than 50 unintended pregnancies and saved the state $1.6 million in public costs [35]. A 2010 study in North Carolina demonstrated that administration of subcutaneous medroxyprogesterone acetate by community pharmacists was not only feasible but showed continuation rates and patient satisfaction comparable to the family planning clinic [36].

Interventions conducted by pharmacists can vary drastically. For instance, the aforementioned “One Key Question^®^” could be a simple question routinely built into existing patient counseling that is routinely offered in community pharmacies nationwide. Accounting for the time spent by adding that question to counseling would be minimal but could have substantial impact if it results in a change in therapy or other more complex intervention. Such interventions have been described in the literature, where pharmacists have billed for targeted medication reviews to offset direct costs associated with the clinical intervention [21,37]. Further, pharmacists have demonstrated success in billing for contraception prescribing in multiple U.S. states [34]. In the case of contraception prescribing, pharmacist fees for service are consistently less than the charge for a physician office visit, which may demonstrate a cost savings. In addition, the increased accessibility of the community pharmacy, frequently offering operating hours longer into the evening and weekend than a traditional physician’s office, may result in fewer absences from work. Convenience may be further improved by offering the visit and dispensing of the contraception product in one location, as compared to the patient needing to travel to the physician’s office and then separately to the pharmacy. Therefore, we believe there to be value to the health care system and its patients when pharmacists are involved not only in contraception provision, but also in broader interventions to support maternal health.

## 5. Accelerating Action

Successful implementation of maternal health services requires pharmacists to be able to practice at the top of their education and training. This includes prescriptive and immunizing authority, the ability to order and interpret relevant tests, and the authority to make and receive patient referrals. Several states in the U.S. have limited regulatory barriers to the extent that pharmacists can leverage their education and training to more fully engage in providing patient care services [38].

Many of the services included as part of a maternal health service are already within the scope of pharmacist practice in all U.S. states. Medication management services are services that pharmacists routinely provide with most encounters focusing on chronic medical conditions. Patients capable of pregnancy may have chronic medical conditions and some of the medications used to treat those conditions are contraindicated in pregnancy. Pharmacists have the expertise needed to create a treatment plan based on the benefits and risks of medication therapy. In some cases, the patient’s chronic medical condition may present a greater risk to the baby if left untreated. Pharmacists can collaborate with the prescriber to tailor the medication regimen to minimize the risk to both mother and child.

At the time of this writing, 21 U.S. jurisdictions have passed statutes or implemented regulations that allow pharmacists to prescribe contraceptives: Arizona, Arkansas, California, Colorado, Delaware, District of Columbia, Hawaii, Idaho, Illinois, Maryland, Minnesota, Nevada, New Hampshire, New Mexico, North Carolina, Oregon, South Carolina, Utah, Vermont, Virginia, and West Virginia [39]. Additionally, several states have broad collaborative practice authority allowing pharmacists to prescribe contraceptives under a collaborative practice agreement with another provider [40].

Approximately 5% of pregnant women use one or more addictive substances, which include alcohol, tobacco and illicit drugs. The use of one of these substances is associated with at least a two times higher incidence of stillbirth [41]. Use of these substances during pregnancy can also lead to preterm birth, low birthweight, birth defects, small head circumference, neonatal abstinence syndrome (NAS), or sudden infant death syndrome (SIDS). Substance use is the leading cause of maternal death in the U.S. Pharmacists are well-trained to provide smoking cessation counseling and education, and in several U.S. states are able to prescribe smoking cessation therapies [42]. In a study including over 1400 participants, researchers showed that pharmacist-provided smoking cessation interventions have quit rates on par with other healthcare professionals [43]. Support to treat addiction represents yet another way that pharmacists can simultaneously support the health of the patient and a potential pregnancy should the patient choose to become pregnant.

While pharmacists in many states in the U.S. are seeing their scope of practice broaden to be able to address such clinical needs, the requisite reimbursement for services may be lacking. Studies have shown barriers to the development of new clinical services in community pharmacies include lack of reimbursement and access to patient medical records as well as workflow and time constraints [44,45,46,47]. Ensuring reimbursement for services provided may help to alleviate challenges related to workflow and time, as it may allow for hiring of additional staff and defraying of other related costs. Demonstration projects have shown that when incentivized, community pharmacists incorporate interventions related to maternal health into their patient care activities [21,37]. Another example of the driver reimbursement can be in the development of such services is the comparison of the availability of pharmacist-prescribed contraception based on statewide policies regarding reimbursement [48,49,50]. Therefore, addressing such barriers as adequate reimbursement for cognitive services and incorporating community pharmacies in health information exchanges are key factors to fully accelerate implementation of provision of maternal health services in community pharmacies [50,51].

## 6. Priorities and the Patient’s Perspective

Different issues may be prioritized at the patient level, such as cost, accessibility, and convenience. Specific to cost, pharmacists may be an affordable source of women’s and maternal health care services. Even though most payers still need to reimburse pharmacists for these services, patients may find out-of-pocket costs for pharmacy services affordable, especially if they have high-deductible plans. As mentioned above, if fees for a patient appointment in a pharmacy are less than similar fees charged by a physician’s office, there is potential for patient-focused cost savings in addition to overall health care cost savings. For instance, concern exists that patients with high deductible health plans and low health literacy may forego preventive and non-preventive health services [52]. If a patient on a high deductible or similar health insurance plan is required to pay for the full cost of the visit, the patient may be highly supportive of a less costly visit with a pharmacist. Additional research suggests that patients on high deductible health plans experience a lower birth rate, but not a significant change in contraception rates [53]. Further research may help to clarify demographics or populations that may find personal economic or other benefits from pharmacist-provided contraception services.

As pharmacists frequently counsel patients about medications, they are also well suited to provide needed counseling for reproductive-age patients. For example, a recent analysis of data collected from women who had a live birth found that less than half with pre-pregnancy diabetes and/or hypertension received recommended counseling prior to conception [54]. Pharmacists can tailor their counseling practices based on patient-specific factors and adapt language according to each patient’s needs. A patient’s receptiveness to needed services may vary based on whether the patient is considering pregnancy in the near future (“contemplator”) or not (“non-contemplator”). For non-contemplators, suggested interventions should be framed as beneficial for their overall health. For those contemplating a pregnancy, emphasizing the benefits of the suggested intervention for pregnancy and infant outcomes may resonate more than with a non-contemplator [55].

Furthermore, there is a need to raise awareness among patients and the general public that that community pharmacists are able and capable to provide maternal health services. One study showed that while over 90% of female patients surveyed at two independent pharmacies had a need for at least one pharmacist-provided preconception health service, only 56% were interested in learning more about such services provided at the community pharmacy and 19% were interested to make an appointment with the pharmacist [56]. Another study showed that less than a third of patients in a state where pharmacists can prescribe hormonal contraception were aware of the option [57]. Community pharmacists must do more to publicize the availability and importance of maternal health services.

## 7. Conclusions

Pharmacists are well-positioned to provide many of the services needed to close the gaps in maternal health care, including but not limited to the prescribing and provision of contraception. The convenience and accessibility of pharmacist-provided services can reduce barriers to care for many individuals. While some studies are available that show the positive impact of pharmacists on specific elements of maternal health care, future studies are needed to show their impact on a more comprehensive suite of maternal health services. Riding on the momentum of recent work to advance contraception prescribing by pharmacists, pilot programs could be conducted to determine the return on investment for pharmacist-provided maternal health services and used to advocate for compensation for these services. Expansion of such services will be essential to ensuring access to care for patients already being served by community pharmacists. We call upon community pharmacists to drive service implementation and expansion to provide proactive maternal health care.

## Figures and Tables

**Figure 1 pharmacy-10-00163-f001:**
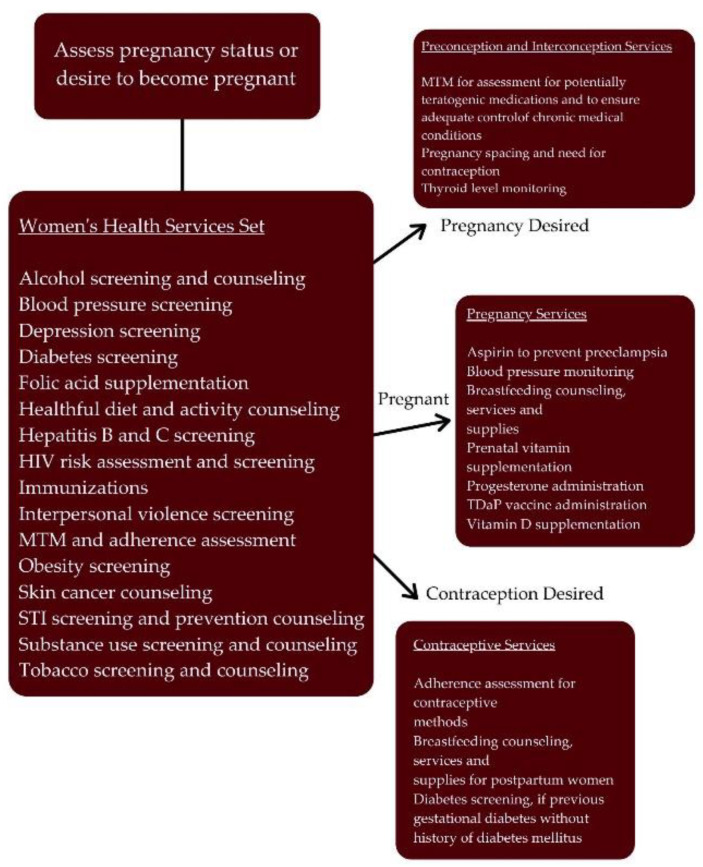
Women’s Health Services Set Flowchart. Reprinted with permission from the National Alliance of State Pharmacy Associations (NASPA).

**Table 1 pharmacy-10-00163-t001:** Women’s Preventive Services Initiative (WPSI) Recommendations Regarding Well-Woman Care for Reproductive-Age Women, 2022.

A. Prevention Services for Reproductive-Age Women		B. Specific Prevention Services During Pregnancy	C. Specific Prevention Services Postpartum
Alcohol screening & counseling Anxiety screening Blood pressure screening Breast cancer risk assessment Cervical cancer screening Contraception & contraceptive care Depression screening Diabetes screening* Folic acid supplementation Healthy diet & activity counseling * Hepatitis B * and C screening HIV risk assessment & screening	HIV preexposure prophylaxis * Immunizations Interpersonal & domestic violence screening Lipid screening * Obesity screening & counseling Skin cancer counseling * STI prevention counseling * STI screening * Substance use screening & assessment Tobacco screening & counseling Tuberculosis screening * Urinary incontinence screening	Services for all reproductive-age women as appropriate (column A) Aspirin therapy to prevent preeclampsia * Bacteriuria screening Breastfeeding counseling, service & supplies Gestational diabetes screening Healthy weight gain counseling Pre-eclampsia screening Pre-eclampsia prevention with low dose aspirin * Rh(D) blood typing	Services for all reproductive-age women as appropriate (column A) Breastfeeding counseling, service & supplies Diabetes screening after gestational diabetes *

* Recommended for selected use. Per WPSI, this information should not be considered as inclusive of all proper treatments or methods of care or as a statement of the standard of care. Comprehensive recommendations for pregnant and postpartum women can be found through the American College of Obstetricians and Gynecologists practice guidelines: https://www.acog.org/clinical?utm_source=redirect&utm_medium=web&utm_campaign=otn (accessed 28 September 2022). To see the latest version of the WPSI Recommendations, go to https://www.womenspreventivehealth.org/ (accessed 28 September 2022).

## Data Availability

Not applicable.

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
