# Peer review of "Beyond Contraception: Pharmacist Roles to Support Maternal Health"

_pharmacy, 2022, doi:10.3390/pharmacy10060163_

Round 1
Reviewer 1 Report
Thank you for this paper. It is unique and offers a great perspective on opportunities for pharmacists to contribute to maternal health. A few comments for consideration:
1. You comment on the lack of primary care providers, which is valid, but is this inclusive of OB-GYNs? Many women of child-bearing age will utilize OB-GYNs or nurse practitioners based in OB-GYN offices for primary care.
2. Line 82: the statement reads, "disease statement management..."; this should be "disease state management"
3. You refer to optimizing disease state management for patients with uncontrolled hypertension and diabetes, yet your flowchart in Figure 1 refers to screening, which are very different approaches for pharmacist-provided care. If you mean to describe pharmacist management of HTN, that would be more comprehensive and likely scheduling of direct patient care visits. You may want to clarify or elaborate further.
4. Line 149: OBRA '90 is simply an offer to counsel and limited to Medicaid patients. Arguably, all patients should receive counseling, but stating it could be routinely built in is likely a stretch.
Reviewer 2 Report
Overall comments: I do think the authors made a good argument for pharmacists providing maternal health services, though do think the organization of content could be improved to flow better. The conclusion statement that “we call upon pharmacists to drive service implementation and expansion to provide proactive maternal health care” is great, but I’m not seeing specifics on how someone would go about this. Some elements are there, like with Figure 1, but what I’d love to see is having the authors take it the next step and point out which services “specifically” pharmacists could do as part of the “maternal health service”. There are also pieces mentioned throughout of the types of services that might be incorporated, but nothing specific and not put together as a cohesive service. For instance, not a lot is mentioned about pharmacists performing screening services specific to what is seen in figure 1, but a huge amount of the care mentioned in that figure has to do with screening.
Not sure if this suggestion would lead to a complete overhaul of the paper, so I’ll leave it to the editor to decide on how important this part is. However, if the spirit of the paper is to encourage pharmacists to start this kind of service, the information on how to do this is missing.
Introduction:
Nice to have the definition of maternal health, but still have some questions:
-Where do individuals usually receive this type of care? PCP’s are mentioned, but what about OBGYN’s, mental health providers, and others?
-How does the US and other countries track maternal health? Chronic health conditions, obesity, maternal morbidity/mortality and infant mortality are mentioned; if these are what is used, that’s fine, but would be nice to mention this.
Would be nice to see more specific information from the references being used as it is helpful to understanding and quantify the connections that the authors are attempting to make. Some examples include:
-If waiting to receive prenatal care affects maternal-fetal outcomes, is there data that shows if individuals are needing to delay prenatal care, and if so what does it say specifically?
-Quantify the PCP shortage, the financial barriers, and how this is specifically impacting or delaying care (how long is it taking for individuals to be seen, are these visits specifically related to maternal care or just care in general?)
-How many individuals experience a financial barrier to care?
-Why are pharmacists “well-positioned” to fill maternal health care gaps? Could mention specifics about their education, or greater accessibility to the public.
After reading through the whole piece, a lot of the specific information missing in the intro, is found later on. Suggest putting more of the discussion points later on into the intro.
Line 45 typo: material, but should be “maternal”
Line 79: would delete the sentence about services listed later since it does not appear to be “listed later” in the paper. Adversely, could mention what the services pharmacists will need to refer for are.
Line 101: love that you mentioned the “one key question”, but would also love to see more information that the cdc has on preconception. They have a large number of resources that could be available to pharmacists such as a developing a reproductive life plan.
Line 112-114: The last sentence doesn’t quite fit with the SDoH argument. Would re-word it to say that “Through this, pharmacists may be able to help address disparities that are often caused by SDoH and that can affect maternal health.” Of course, make sure that lines up with the reference being used for this sentence, as I’m wondering if that reference only refers to “geography or race/ethnicity” instead of the large number of factors that contribute to SDoH.
Line 169-171: Is there a reference for the sentence about states that have reduced regulatory barriers for pharmacists? Would also like to see specifics about what regulatory barriers that are being referred to.
Line 186: Change the tense of that sentence to match the others. Instead of “pharmacists will” just say “pharmacists collaborate…” or “pharmacists can collaborate…”
References: Please go through and correct some of the references as news articles are occasionally cited instead of the primary literature. Examples I’m seeing include reference 17, reference 21, and reference 33 (cdc may have a media site, but the “implementation kit” probably has authors that should be credited).
